

# To flea or not to flea: survey of UK companion animal ectoparasiticide usage and activities affecting pathways to the environment

Rosemary Perkins and Dave Goulson

School of Life Sciences, University of Sussex, Brighton, United Kingdom

## ABSTRACT

Due to the convenience and efficacy of modern ectoparasiticides, routine prophylactic use has become commonplace for dogs and cats. However, the environmental consequences of this large-scale use are not well-understood, and multiple potential pathways for ectoparasiticides to the environment exist. Of particular concern is the potential for topically applied ectoparasiticides to pass to waterways, both down-the-drain via wastewater treatment plants and directly through swimming. In this online cross-sectional survey of 1,009 UK cat and dog owners, we investigated ectoparasiticide usage and the frequency of activities that are likely to result in transfer of the active substance to the environment, with a focus on pathways to waterways. A total of 603 dog owners and 406 cat owners completed the survey. Amongst dog and cat owners, 86.1% and 91.1% had administered an ectoparasiticide treatment to their pet in the preceding 12 months. Imidacloprid was the most frequently administered ectoparasiticide in both cats and dogs, followed by fluralaner in dogs and fipronil in cats. Eighty-four percent of owners who applied topical ectoparasiticides to their dog said they were aware of product warnings regarding swimming and bathing after application. Spot-on treated dogs were reported to swim significantly less frequently than non spot-on treated dogs ($p = 0.007$); however, 36.2% were reported to swim at least monthly. Similarly, significant differences were found in bathing frequency between spot-on treated and non spot-on treated dogs, with treated dogs less likely to be bathed at frequent (weekly) intervals, however 54.6% were reported to be bathed at least monthly. Washing of bedding was unaffected by ectoparasiticide treatment, and 87.8% of dog owners and 69.1% of cat-owners reported washing their pet's bedding at least every 3 months, suggesting that residue washoff from bedding may be occurring for most topically treated animals. Results suggest that transfer of ectoparasiticides to the environment is likely to occur for many of the millions of animals treated annually in the UK, with imidacloprid spot-on treated dogs estimated to swim, be bathed and have their bedding washed over 3.3 million, 5 million and 6.3 million times per year, respectively.

Corresponding author
Rosemary Perkins,
rp442@sussex.ac.uk

## INTRODUCTION

The United Kingdom is often described as a nation of animal-lovers. Indeed, 52% of UK adults own a pet, and there are an estimated 10.2 million dogs and 11.1 million cats in the UK (*People's Dispensary for Sick Animals, 2022*). Many owners consider their pets to be part of the family, going to great lengths to ensure that their companions enjoy a happy and healthy life. Ectoparasites such as fleas and ticks cause irritation and discomfort to pets and may be vectors for disease, and their effective control is a concern for many pet owners. The introduction in the 1990s of potent, long acting spot-on ectoparasiticides such as fipronil and imidacloprid enabled more effective treatment of ectoparasite infestations (*Marsella, 1999*; *Rust, 2005*), and—due to their ease and convenience—ushered in an era of routine prophylactic ectoparasite control, with many pets receiving continuous year-round ectoparasiticide treatment (*British Veterinary Association, 2021*).

Sales in the UK for companion animal ectoparasiticides have surged in subsequent decades, with annual imidacloprid sales increasing over forty-fold from approximately 103 kg in 1997 to 4,170 kg in 2017, and fipronil sales increased from approximately 566 kg to 1,584 kg during that time (Veterinary Medicines Directorate, 2022, unpublished data). Following the introduction of newer classes of ectoparasiticides such as isoxazolines, a substantial proportion of the market share has shifted towards these products, although recent data suggests that fipronil and imidacloprid sales still constitute the majority of the pet ectoparasiticide market in the UK (*Wells & Collins, 2022*; *Cooper et al., 2020*).

There has been growing concern in recent years over the environmental consequences of this widespread ectoparasiticide usage. Both fipronil and imidacloprid have been restricted for agricultural use in the UK due to concerns regarding their impact on non-target invertebrates. Fipronil's approval for use as a plant protection product ended in 2017 (*European Commission, 2019a*) and the outdoor use of imidacloprid was banned in 2018 (*European Food Safety Authority, 2018a*). At present no plant protection products containing fipronil or imidacloprid are registered for use in the UK (*Health and Safety Executive, 2022*). However widespread, largely unrestricted veterinary use in companion animals continues. Of particular concern is evidence that topically applied ectoparasiticides are passing down-the-drain (DTD) from households, persisting during wastewater treatment, and entering waterways *via* wastewater treatment facilities. Following the ubiquitous detection of fipronil and imidacloprid in Californian wastewater effluent, companion animal ectoparasiticides were identified as a potentially important source of waterway pollution (*Sadaria et al., 2017*). The relatively low variability in daily per capita load demonstrated in this study suggests many ubiquitous sources of the measured pollution—consistent with DTD transfer from a large number of treated pets. *Teerlink, Hernandez & Budd (2017)* subsequently confirmed DTD transport of fipronil and its toxic degradates (fiproles) through bathing of spot-on treated dogs —demonstrating residues in washoff for at least 28 days after application. *Webb & Zhi (2021)* found that imidacloprid levels in an effluent-dominated stream in Iowa were driven year-round by wastewater effluent point-sources, whereas neonicotinoids with no veterinary use originated primarily from upstream non-point sources. This study estimated that if 1% of

the imidacloprid applied to pets passed DTD this would account for a large proportion of the measured wastewater pollution. In England, widespread waterway pollution with fiproles and imidacloprid has been demonstrated in urban rivers (*Richardson et al., 2022*), with household veterinary parasiticides in wastewater effluent implicated as a source of this pollution (*Perkins et al., 2020*; *Robinson et al., 2022*). Subsequently, the near-ubiquitous detection of fipronil and imidacloprid in wastewater effluent from multiple treatment plants following their inclusion in the 3rd UKWIR (UK Water Industry Research) Chemical Investigation Programme (CIP3) has provided further evidence to support these concerns (*UK Water Industry Research 2023*).

Companion animal parasiticides are available in a variety of formulations including spot-ons, tablets, collars and injections, and multiple pathways for active substances to the environment exist in addition to bathing. Cutaneously distributed, topically applied ectoparasiticides such as fipronil and imidacloprid are disseminated to the environment through shed hair and skin, and through direct transfer (*Dyk et al., 2012*; *Jacobs et al., 2001*). Washing of pet bedding or other in-contact textiles and washing of owner hands are likely additional DTD pathways to waterways for these compounds. They are also likely to enter waterways directly when treated animals swim, through leaching of applied product. Some systemic absorption of topically applied, cutaneously distributed parasiticides occurs, with both fipronil and imidacloprid being demonstrated in the blood of animals following topical application (*Craig et al., 2005*; *Arisov, Indyuhova & Arisova, 2019*). Following systemic absorption, varying levels of fipronil and imidacloprid—and their associated metabolites—have been demonstrated in urine and stool (*Gupta & Anadón, 2018*; *Cravedi et al., 2013*; *Gupta et al., 2014*). However, a low proportion (11%) of topically applied fipronil appears to be systemically absorbed (*National Office of Animal Health, 2022*), suggesting that excreta may be an environmental exposure pathway of lesser significance.

Orally or topically administered isoxazolines are systemically absorbed, and fluralaner is primarily excreted unchanged in the stool (*European Medicines Agency , 2013*; *European Medicines Agency , 2016*); however, accumulation in the skin and hair occurs (*European Medicines Agency, 2016*), suggesting that dissemination to the environment may also occur through skin and hair for this parasiticide class. Further studies are required to quantify the load entering the environment through various pathways for the different parasiticides—including studies providing reliable emissions fractions for the routes and activities described above, and studies investigating the frequency of emitting activities. This study aims to shed light on the frequency of activities that are likely to lead to transfer of ectoparasiticides from pets to the environment, with a focus on DTD and direct pathways to waterways, including bathing of dogs (*Teerlink, Hernandez & Budd, 2017*), washing of their bedding (*Jacobs et al., 2001*) and swimming (*Diepens et al., 2023*).

Under the two-tier VICH (International Cooperation on Harmonisation of Technical Requirements for Registration of Veterinary Medicine Products) environmental impact assessment that veterinary medicines currently undergo in the European Union, the UK, Japan, the United States, and various other nations, the environmental exposure of veterinary medicines intended for use in non-food animals is assumed to be low in the Phase I assessment, and no Phase II assessment, providing data on their environmental fate

and impact, is required (*European Medicines Agency , 2000*). Following the publication of a reflection paper on risk mitigation measures related to environmental risk assessments (*European Medicines Agency , 2011*), a recommendation was made to include a minimum time interval for which dogs treated topically with an active substance with known aquatic toxicity should not be allowed to enter surface water. The default recommendation for this period is 48 h, after which it is assumed that release of active substance from fur will be negligible (although it should be noted that no supporting data were provided to support this assumption). The risk assessment process in the UK and European Union, or other VICH countries does not consider other potential routes to the environment for environmentally toxic active substances used in companion animals including DTD transfer, or soil pollution *via* excreta. It is worth mentioning that Phase II ecotoxicity studies were carried out for Stronghold® (Zoetis, containing selamectin) and Advocate® (Elanco, containing imidacloprid and moxidectin). It is beyond the scope of this paper to discuss these, however it should be noted that the standard ecotoxicity test species considered in these assessments, the water flea *Daphnia magna*, displays an unusually high tolerance towards neonicotinoids, including imidacloprid (*Morrissey et al., 2015*; *Li et al., 2021*).

This study investigates ectoparasiticide usage amongst dog and cat owners in the UK and the frequency of activities that are likely to result in transfer of active substance to the environment, with a focus on pathways to waterways. It is intended as a first step in quantifying the different pathways for pet ectoparasiticides to the environment, and to aid in the estimation of the contribution of these products to waterway pollution. Further studies are required to establish emissions fractions from the various routes included in the study, such as washing of bedding and swimming, and to estimate the resultant load of DTD and direct emissions to waterways from the pet population.

## MATERIALS AND METHODS

A multiple choice and short answer online survey was designed to assess the use of companion animal ectoparasiticides and the frequency of activities likely to affect their transmission to the environment.

Using the method outlined by *Daniel (1999)*, a minimal sample size of 385 for both cat and dog owners was calculated from the following formula:

$$n = z^2(1-p)p/e^2$$

where $n$ is the estimated sample size required for the desired precision and confidence; $z$ is the two-tailed value of the standardized normal deviation associated with the desired level of confidence of 0.95; $p$ is the preliminary estimate of population proportion which maximises variance and sample size (*Bartlett, Kotrlik & Higgins, 2001*), 0.5; and e is the desired margin of error of 0.05.

A pilot survey was distributed online to 155 respondents and questions adjusted as required to improve the survey. The survey, titled "Evaluating the use of flea treatments in the UK" (File S1) was available online *via* Qualtrics from June to July 2020, with

participants recruited through social media outlets such as Facebook and Twitter. A broad range of Facebook and Twitter pages including the authors' feeds, general pet interest and pet training organisations were used to distribute the survey. All data collected were anonymous. All respondents were over 18 years of age, lived in the UK and answered questions pertaining to one cat or dog that had been in their household for at least one year, unless it was a puppy or kitten.

The survey included questions to gather demographic information about the pet owner, ectoparasiticide product usage, and the frequency of activities that are likely to result in transfer of parasiticide ingredients to the environment from treated pets.

Most questions were multiple choice; however, some short answer options were included. When asked to specify ectoparasiticide products used over the previous 12 months, respondents were provided with a list of the most commonly used products as well as an 'other' option in which a product could be specified. Products were considered 'alternative' if they were not listed in the VMD's Product Information Database. Where more than one active ingredient was present in a product, the main flea adulticide, determined by mass, was reported, Further information on active ingredients is provided in File S2. Spot-on products containing fipronil, with the prefix 'Fip' (*e.g.*, Fipnil, Fiprene, Fiproclear) were classified as 'Fipronil generics'. Ectoparasiticide products were referred to as 'flea products' in the questionnaire as all authorised products included in the survey have efficacy against fleas, but not all have efficacy against ticks. Pets are referred to as 'spot-on treated' or 'collar-treated' if their owner indicated that the main ectoparasiticide product applied to their dog over the preceding 12 months was a spot-on product or a collar. Unless otherwise stated, respondents who indicated 'Don't know' to a question were excluded from the analysis for that question.

The survey received ethical approval from the Sciences & Technology Cross-Schools Research Ethics Committee (Project reference number ER/AT459/3). Informed consent was obtained, and the participants were made aware that participation is voluntary and that they were free to withdraw at any point. All collected data was kept anonymised and confidential.

A total of 1,021 questionnaires were submitted, of which 1,009 were complete enough to include for statistical analysis. Survey responses were analysed and descriptive statistics, including proportions and confidence intervals (CI), were used to assess categorical data. The Chi-square test was used to assess the relationship between variables of interest.

All analysis was performed and all figures created using R Studio software (version 1.2.5042-1; *RStudio Team, 2020*).

## RESULTS

Survey participants comprised 406/1009 (40.3%, 95% CI [37.2–43.3]) cat owners and 603/1009 (59.7%, 95% CI [56.7–62.8]) dog owners. The demographic characteristics of respondents are presented in Table S1.

**Table 1** Reported swimming frequency in all dogs, spot-on treated dogs, non spot-on treated dogs and collar treated dogs.

| Swimming frequency | All dogs | | Spot-on treated | | Non spot-on treated | | Collar treated | |
|---|---|---|---|---|---|---|---|---|
| | *n* | % | *n* | % | *n* | % | *n* | % |
| Every week or more | 142 | 23.8 | 40 | 17.9 | 102 | 27.4 | 10 | 37 |
| Every 2 weeks | 57 | 9.6 | 16 | 7.1 | 41 | 11 | 2 | 7.4 |
| Every month | 67 | 11.2 | 25 | 11.2 | 42 | 11.3 | 3 | 11.1 |
| Every 3 months | 40 | 6.7 | 13 | 5.8 | 27 | 7.3 | 2 | 7.4 |
| Every 6 months | 35 | 5.9 | 14 | 6.2 | 21 | 5.6 | 2 | 7.4 |
| Once a year or less | 255 | 42.8 | 116 | 51.8 | 139 | 37.4 | 8 | 29.6 |
| Total | 596 | 100 | 224 | 100 | 372 | 100 | 27 | 100 |

**Notes.**

*n*, number of responses to the question "How often does your dog enter into or swim in lakes or rivers?".; %, percentage of responses.

## Use of ectoparasiticides

Amongst dog and cat owners, 519/603 (86.1%, 95% CI [83.0–88.7]) and 370/406 (91.1%, 95% CI [87.8–93.6]) had administered an ectoparasiticide treatment to their pet in the preceding 12 months. 428/519 (82%, 95% CI [78.9–85.6]%) of dog owners and 294/370 (79.5%, 95% CI [74.9–83.4]) of cat owners indicated that prevention of fleas (and/or other parasites such as ticks) was their primary reason for administering an ectoparasiticide, with the remaining respondents indicating that treatment was primarily in response to flea or tick infestation. Figures 1 and 2 provide a summary of the most commonly used ectoparasiticides products in dogs and cats as reported by owners.

Amongst dog owners that had administered an ectoparasiticide product in the preceding 12 months, Bravecto tablets were most commonly used, with 26% (128/492, 95% CI [22.2–30.2]) of respondents indicating that this was the main product used. This was followed by Advocate (93/492; 18.9%, 95% CI [15.6–22.7]), NexGard Spectra (66/492; 13.4%, 95% CI [10.6–16.8]) and Frontline (52/492; 10.6%, 95% CI [8.6–13.7]). Grouped by active ingredient, products containing imidacloprid were most often used (159/492; 32.3%, 95% CI [28.2–36.7]), followed by fluralaner (134/492; 27.2%, 95% CI [23.3–31.4]), afoxalaner (81/492; 16.5%, 95% CI [13.4–20.1]) and fipronil (78/492; 15.9%, 95% CI [12.8–19.5]).

Advocate was the product most used by cat owners (59/353; 16.7%, 95% CI [13.1–21.1]), followed by Frontline (58/353; 16.4%, 95% CI [12.8–20.8]), Bravecto (52/353; 14.7%, 95% CI [11.3–19.0]) and Advantage (50/353; 14.1%, 95% CI [10.8–18.3]). Imidacloprid was the most frequently administered active ingredient (121/353; 34.3%, 95% CI [29.4–39.5]), followed by fipronil (110/353; 31.2%, 95% CI [26.4–36.3]), fluralaner (52/353; 14.7%, 95% CI [11.3–19.0]) and selamectin (43/353; 12.2%, 95% CI [9.0–16.2]).

Ectoparasiticide products were most often administered to dogs in tablet form, with 48% (236/492, 95% CI [43.5–52.5]) of dogs receiving treatment orally, whilst 46.3% (228/492, 95% CI [41.9–50.9]) were treated with spot-ons and 28/492 (5.6%, 95% CI [3.9–8.2]) with collars. However, cats were far more likely to be treated with spot-ons, with 92% (325/353, 95% CI [88.6–94.6]) of treatments being administered by this route, compared to 15/353

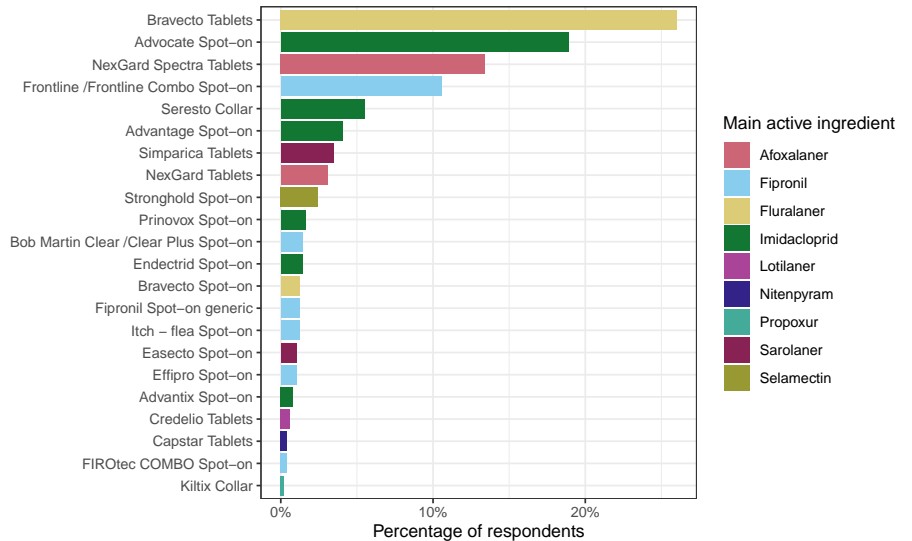

**Figure 1 Dog ectoparasiticides products.** Responses to the question: "During the last 12 months, what is the main flea product used on your dog?", $n = 492$, respondents that had not treated their dog in that time ($n = 83$) or applied an unrecognised/alternative treatment ($n = 14$) are not included. Main active ingredient = main flea adulticide, determined by mass. Note that active ingredients beyond the main flea adulticide are not displayed here (for example, flumethrin in Seresto collars). Further information on products and active ingredients is provided in File S2.

(4.2%, 95% CI [2.5–7.0]) receiving treatment orally, 8/353 (2.2%, 95% CI [1.1–4.6]) being treated with collars and 5/353 (1.4%, 95% CI [0.5–3.4]) receiving treatment *via* injection. Figures 3 and 4 summarise the administration routes and main active ingredients of dog and cat ectoparasiticides products identified in survey responses.

### Purchasing behaviour

Thirty-four percent (345/1004, 95% CI [31.4–37.4]) of pet owners stated that they had a regular subscription for year-round ectoparasiticides. Categorised by species, 36.1% (217/600, 95% CI [32.3–40.1]) of dog owners and 31.7% (128/404, 95% CI [27.2–36.5]) of cat owners subscribed to a regular ectoparasiticide plan.

Dog owners were more likely than cat owners to purchase ectoparasiticides from a veterinary practice, with 369/519 (71.1%, 95% CI [67.0–74.9]) reporting having done so, whilst 95/519 (18.3%, 95% CI [15.1–22.0]) purchased treatments online and 55/519 (10.6%, 95% CI [8.1–13.6]) purchased treatments from a petshop, supermarket or elsewhere.

Amongst cat owners, 63.8% (236/370, 95% CI [58.6–68.6]) indicated that they usually purchased ectoparasiticides from a veterinary practice, whilst 23.2% (86/370, 95% CI [19.1–28.0]) purchased products online and 13% (48/370, 95% CI [9.8–16.9]) chose to purchase from a petshop, supermarket or elsewhere.

### Awareness of product warnings

Dogs owners were asked: "*If you have applied a spot-on flea product or flea collar to your dog, are you aware of product warnings regarding swimming or bathing after flea*

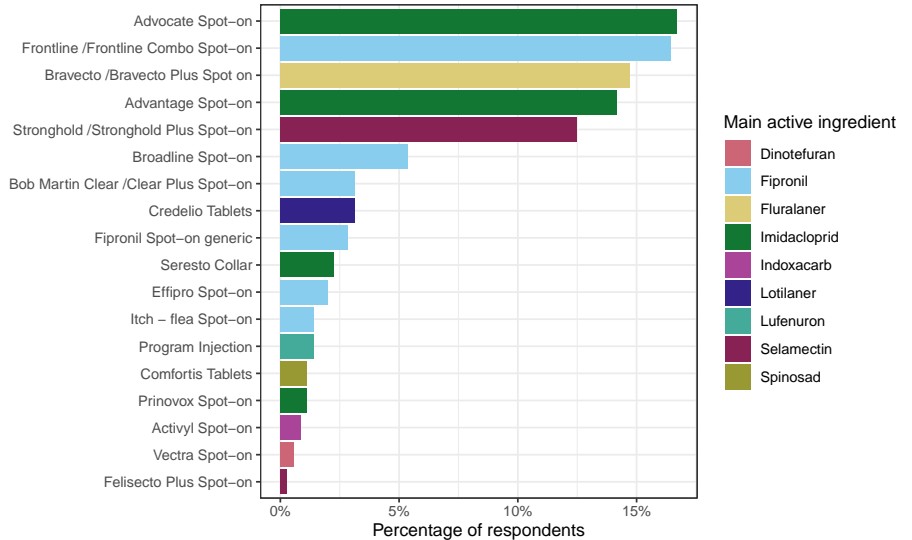

**Figure 2  Cat ectoparasiticide products.** Responses to the question: "During the last 12 months, what is the main flea product used on your cat?". $n = 353$, respondents that had not treated their cat in that time ($n = 36$) or applied an unrecognised/ alternative treatment ($n = 7$) are not included. Main active ingredient = main flea adulticide ingredient, determined by mass. Note that active ingredients beyond the main flea adulticide are not displayed here (for example, flumethrin in Seresto collars). Further information on number of respondents and active ingredients is provided in File S2.

*treatment?"*, to which 84.4% (232/275, 95% CI [79.4–88.3]) responded "Yes". Respondents who indicated that they were aware of product warnings regarding swimming and bathing were asked if they followed these warnings, to which 97.3% (220/226, 95% CI [94.0–98.9]) answered in the affirmative.

## Veterinary advice

Survey participants were asked to indicate what advice (if any) they had received from their veterinarian regarding ectoparasiticide control. A summary of responses is provided in Table S1.

The majority (81.3%; 794/977, 95% CI [78.6–83.6]) of pet owners reported that they were advised by their veterinarian to provide prophylactic flea/tick treatment throughout the year. Eleven percent (108/977, 95% CI [9.1–13.2]) received no advice, and the remaining respondents (7.6%; 75/977, 95% CI [6.1–9.6]) indicated that they were given risk-based advice such as treating only in warmer months or were advised only to treat in response to an infestation.

## Activities affecting transfer to the environment
### *Swimming*

Table 1 provides a summary of reported swimming behaviour in dogs.

Overall, 44.6% (266/596, 95% CI [40.6–48.7]) of dogs were reported to swim at least monthly. Spot-on treated dogs swam less frequently than non spot-on treated dogs ($X^2_{(5)} = 15.87$, $p = 0.007$), with 36.2% (81/224, 95% CI [30.0–42.8]) reported to swim at least
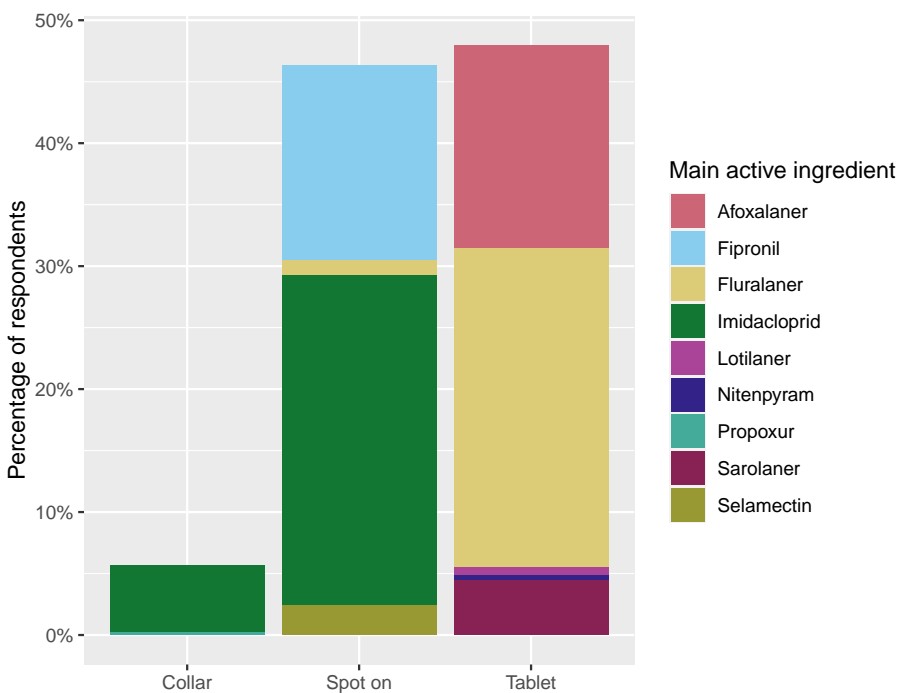

**Figure 3** **Dog ectoparasiticide administration routes.** Responses to the question: "During the last 12 months, what is the main flea product used on your dog?" (Fig. 1), represented by administration route and main active ingredient. $n = 492$, respondents that had not treated their dog in that time ($n = 83$) or applied an unrecognised/alternative treatment ($n = 14$) are not included. Main active ingredient = main flea adulticide, determined by mass. Note that active ingredients beyond the main flea adulticide are not displayed here (for example, flumethrin in Seresto collars). Further information on products and active ingredients is provided in File S2.

monthly, and 17.9% (40/224, 95% CI [13.2–23.6]) swimming every week or more. 49.7% (185/372, 95% CI [44.5–54.9]) of non-spot-on treated dogs were reported to swim at least monthly. Amongst collar treated dogs, 55% (15/27, 95% CI [35.6–74.0]) were reported to swim at least monthly, with 37% (10/27, 95% CI [20.0–57.5]) swimming every week or more, however the small sample size for the collar-treated group means that these results should be considered indicative only.

### Bathing

Dog owners were asked to indicate how frequently they bathed their dog. Table 2 provides a summary of results. Amongst all dogs, 52.9% (314/594, 95% CI [48.8–56.9]) were bathed at least monthly, with 12% (71/594, 95% CI [9.5–14.9]) being bathed weekly or more. Significant differences in bathing frequency were found between spot-on treated and non spot-on treated dogs ($X^2_{(5)} = 20.96$, p = 8e−4), with spot-on treated dogs less likely to be bathed weekly (8.1%; 18/223, 95% CI [5.0–12.7]), but with a higher proportion being bathed at least monthly (56.1%; 125/223, 95% CI [49.3–62.6]). A relatively high proportion of collar-treated dogs were bathed weekly or more often (28.6%; 8/28, 95% CI [14.0–48.9]);

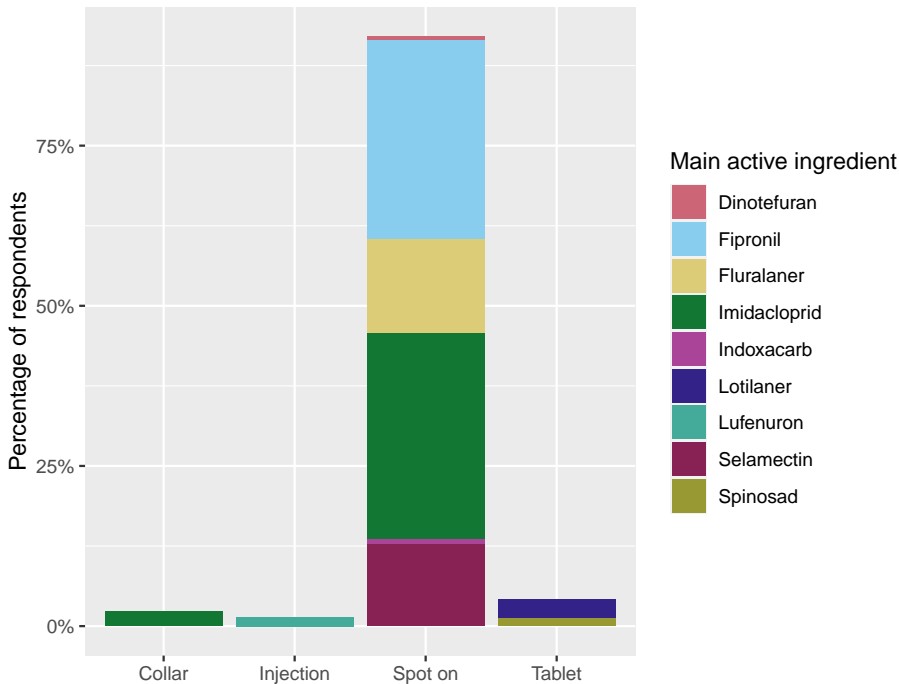

**Figure 4 Cat ectoparasiticides administration routes.** Responses to the question: "During the last 12 months, what is the main flea product used on your cat?" (Fig. 2), represented by administration route and main active ingredient. $n = 353$, respondents that had not treated their cat in that time ($n = 36$) or applied an unrecognised/ alternative treatment ($n = 7$) are not included. Main active ingredient = main flea adulticide ingredient, determined by mass. Note that active ingredients beyond the main flea adulticide are not displayed here (for example, flumethrin in Seresto collars). Further information on number of respondents and active ingredients is provided in File S2.

**Table 2 Reported bathing frequency in all dogs, spot-on treated dogs, non spot-on treated dogs and collar treated dogs.**

| Bathing frequency | All dogs | | Spot-on treated | | Non spot-on treated | | Collar treated | |
|---|---|---|---|---|---|---|---|---|
| | *n* | % | *n* | % | *n* | % | *n* | % |
| Every week or more | 71 | 12 | 18 | 8.1 | 53 | 14.3 | 8 | 28.6 |
| Every 2 weeks | 62 | 10.4 | 24 | 10.8 | 38 | 10.2 | 3 | 10.7 |
| Every month | 181 | 30.5 | 83 | 37.2 | 98 | 26.4 | 4 | 14.3 |
| Every 3 months | 130 | 21.9 | 57 | 25.6 | 73 | 19.7 | 4 | 14.3 |
| Every 6 months | 67 | 11.3 | 22 | 9.9 | 45 | 12.1 | 3 | 10.7 |
| Once a year or less | 83 | 14 | 19 | 8.5 | 64 | 17.3 | 6 | 21.4 |
| Total | 594 | 100 | 223 | 100 | 371 | 100 | 28 | 100 |

**Notes.**
*n*, number of responses to the question "How often do you wash or hose down your dog?".; %, percentage of responses.

however, the small sample size for the collar-treated group means that these results are indicative only.

**Table 3  Responses to the question "What is your dog/cat's most regular sleeping place? Select one".**

| Sleeping location | Dogs | | Cats | |
|---|---|---|---|---|
| | *n* | % | *n* | % |
| In their own bed | 249 | 41.3 | 86 | 21.2 |
| On the furniture | 113 | 18.7 | 154 | 37.9 |
| On the floor | 51 | 8.5 | 29 | 7.1 |
| On a family member's bed | 123 | 20.4 | 122 | 30.0 |
| In a crate | 59 | 9.8 | 0 | 0.0 |
| Outdoors kennel/building | 7 | 1.2 | 10 | 2.5 |
| Other/don't know | 1 | 0.2 | 5 | 1.2 |
| Total | 603 | 100 | 406 | 100 |

**Notes.**

*n*, number of responses.; %, percentage of respondents.

When asked to indicate where their dog was usually bathed or washed down, 61.6% (352/571, 95% CI [57.5–65.6]) indicated that this occurred in a bath, sink or shower, whilst 15.4% (88/571, 95% CI [12.6–18.7]) of dogs were usually bathed at a groomers, and 22.9% (131/571, 95% CI [19.6–26.7]) were washed outdoors—indicating that washoff usually goes down-the-drain in at least 77.1% (440/571, 95% CI [73.3–80.4]) of bathed dogs.

### Bed washing

Dog and cat owners were asked to provide information on where their pets slept, and how often they washed their pet's bedding.

Both cats and dogs were found to sleep in a variety of locations, with dog owners indicating an average of 2.3 (95% CI [2.2–2.4]) different sleeping locations, and cat owners selecting an average of 2.9 locations (95% CI [2.8–3.0]). 41.3% (249/603, 95% CI [37.3–45.3]) of dogs slept primarily in their own beds (Table 3), whilst 20.4% (123/603, 95% CI [17.3–23.9]) slept mainly on a family member's bed, with the remaining owners indicating that dogs slept on furniture, flooring, in a crate, outdoor kennel, or elsewhere. By comparison, cats were more likely to sleep on the furniture (37.9%; 154/406, 95% CI [33.2–42.9]) or a family member's bed (30%; 122/406, 95% CI [25.7–34.8]) than their own bed (21.2%; 86/406, 95% CI [17.3–25.5]). It should be noted that some parasiticide products warn against allowing pets to sleep in the same bed as owners, particularly children, whilst being treated with the product (Veterinary Medicines Directorate, 2022, unpublished data).

A summary of bed washing frequency is provided in Table 4. 67.2% (395/588, 95% CI [63.2–70.9]) of dog owners reported that they washed their dog's bedding at least monthly, and 87.8% (516/588, 95% CI [84.8–90.2]) indicated that they washed their bedding at least every 3 months.

Cat owners reported less frequent washing of bedding with 55.6% (205/369, 95% CI [50.3–60.7]) stating that they washed their cat's bedding at least monthly and 69.1% (255/369, 95% CI [64.1–73.7]) stating that they did so at least every 3 months—likely the result of cats being more likely to sleep on household furniture or family bedding.

**Table 4   Reported bed washing frequency in dogs and cats.**

| Bed washing frequency | Dogs | | Cats | |
|---|---|---|---|---|
| | *n* | % | *n* | % |
| Every week or more | 90 | 15.3 | 42 | 11.4 |
| Every 2 weeks | 127 | 21.6 | 65 | 17.6 |
| Every month | 178 | 30.3 | 98 | 26.6 |
| Every 3 months | 121 | 20.6 | 50 | 13.6 |
| Every 6 months | 29 | 4.9 | 35 | 9.5 |
| Once a year or less | 43 | 7.3 | 79 | 21.4 |
| Total | 588 | 100 | 369 | 100 |

Notes.

*n*, number of responses to the question "How often do you wash your dog/cat's bedding?"; %,  percentage of respondents.

No significant associations were found between bed washing frequency and any route of administration of ectoparasiticide treatment in dogs or cats ($p > 0.05$).

### Disposal of animal waste

Twenty nine percent (117/403, 95% CI [24.7–33.8]) of cat owners indicated that their cat never used a litter tray, and 30.5% (123/403, 95% CI [26.1–35.3]) indicated that their cat always used a litter tray, with the remaining respondents (40.4%, 163/403, 95% CI [35.6–45.4]) indicating that their cat sometimes used a litter tray. Cat owners were asked "If your cat uses a litter tray, how do you dispose of the waste litter? Select all that apply". Eighty seven percent (279/307, 95% CI [93.9–98.3]) indicated that they threw waste litter in the bin, 10.1% (31/307, 95% CI [7.1–14.2]) indicated that they flushed it down the toilet, and 5.9% (18/307, 95% CI [3.6–9.3]) indicated "Other/don't know".

When asked how often they picked up their dog's stool, 65.2% (392/601, 95% CI [61.2–69.0]) of owners stated that they always did so and 1.2% (7/601, 95% CI [0.5–2.5]) indicated that they never did, with the rest (33.6%, 202/601, 95% CI [29.9–37.6]) indicating that they sometimes or usually did so.

When asked how they disposed of their dog's poo, 94.7% (570/602, 95% CI [92.5–96.3]) indicated that their main method of disposal was in a waste bin. Further detail is provided in Table S5.

### Estimating the scale of polluting activities

Product sales data from the VMD indicates that in 2018 approximately 2,352.5 kg of imidacloprid (referring to active substance only) was sold in dog spot-on products (*VMD, 2022*). Given that the monthly imidacloprid spot-on dose for an average sized dog of 15 kg is 250 mg (*National Office of Animal Health, 2022a*; *National Office of Animal Health, 2022*), we estimate that around 9.4 million imidacloprid spot-on doses were applied to dogs in that year.

Results from this survey indicate that over a third of spot-on treated dogs are likely to swim in waterways at least once a month, over half are likely to be bathed at least once a month, and over two-thirds have their bedding washed at least monthly. Given the estimated number of imidacloprid spot-ons applied to dogs in the UK each year, this

suggests dogs treated with these products are likely to swim, be bathed or have their bedding washed at least 3.3 million, 5 million and 6.3 million times per year, respectively.

## DISCUSSION

### Ectoparasiticide usage

These results help to elucidate the nature and scale of ectoparasiticide usage in UK pets, and the frequency of activities that are likely to result in transfer of ectoparasiticides to the aquatic environment. As with any online cross-sectional survey, there is potential for response bias if individuals with greater interest in the topic were more likely to respond. The limitations of retrospective questionnaires should also be considered, as these depend on respondents' recollections.

Consistent with the findings of previous research (*Wells & Collins, 2022*; *Cooper et al., 2020*), results suggest that whilst newer ectoparasiticides such as fluralaner and afoxalaner are popular, with Bravecto (containing fluralaner) being the most used product in dogs, sales of older products containing imidacloprid and fipronil remained high in 2019/2020 - with imidacloprid being the most commonly administered ectoparasiticide ingredient in both cats and dogs.

Grouped by administration route, tablets were marginally more popular amongst dog-owners than spot-ons, whereas cats were far more likely to receive ectoparasiticides treatment in the form of spot-ons—presumably the result of cats being considered difficult to medicate orally at home (*Savolainen, 2020*). The administration route and pharmacokinetic distribution of parasiticide products are likely to have implications for environmental exposure, and further research is needed to investigate environmental exposure pathways for different classes of parasiticides.

There is growing awareness of environmental concerns surrounding large scale ectoparasiticide usage, with much discussion on appropriate usage of these products within the veterinary profession (*Little & Boxall, 2020*; *Whitehead, Perkins & Goulson, 2021*). In the UK, the British Veterinary Association published a policy position statement on responsible parasiticide usage in cats and dogs in 2021 (*British Veterinary Association, 2021*) which highlights some of the concerns and knowledge gaps surrounding this subject, and encourages a risk-based approach to parasiticides which avoids blanket treatment of pets. Instead, risk assessments on individual animals which consider various factors such as lifestyle and season are encouraged, and veterinary businesses are recommended not to have blanket treatment policies in place, instead empowering individual vets to have those conversations with their client. Tools using electronic health data to provide an indication of risk, such as the University of Liverpool's 'Flea activity index' which maps flea activity across the UK (*SAVSNET, 2023*), support evidence-based decision-making that may enable a more integrated approach to the control of parasites (*Otranto & Wall, 2008*).

Recent publications have recommended additional measures to address the overuse of pet parasiticides, including better regulation of product advertising, raising awareness of potential environmental hazards associated with their use through the provision of improved product literature (*European Medicines Agency , 2023*), stricter environmental

assessments prior to regulatory approval and reclassification to prescription-only status (POM-V or POM-VPS; *Preston-Allen et al., 2023*).

Most survey respondents were advised by veterinary professionals to provide year-round preventative ectoparasiticide treatment for their pet (81.3%; 794/977) or were given no advice regarding ectoparasites (11.1%; 108/977) —suggesting that little in the way of individual risk assessments were being performed at the time of this survey (June 2020). The prevalence of veterinary practice healthcare plans, many of which promote blanket year-round parasite control in pets (*Bagster & Elsheikha, 2022*), may be affecting the freedom of veterinarians to have frank discussions with owners about the risks, benefits and uncertainties surrounding parasite control in pets.

## Datasheet guidelines

Product datasheets for spot-on and collar products provide guidelines regarding bathing of treated animals and entry into watercourses.

Guidelines on swimming following application of spot-on parasiticides have largely reflected the 2011 EMA risk mitigation guidelines (*European Medicines Agency , 2011*), providing minimum time intervals before treated dogs can swim, to mitigate against potential environmental harm. A default value of 48 h is advised, however little research has been done to investigate whether these guidelines provide adequate protection. It is well-established that cutaneously distributed ectoparasiticides are present in the skin and coat of treated animals for the entire period of efficacy—usually at least 4 weeks (*European Medicines Agency, 2009*; *Dyk et al., 2012*), suggesting that ectoparasiticides may be entering waterways well beyond the minimum time interval indicated by datasheet guidance. Products containing fipronil, imidacloprid (without additional moxidectin), dinotefuran, permethrin, pyriproxifen and indoxacarb state that dogs should not be allowed to swim in watercourses for 48 h after application to avoid adverse effects on aquatic organisms (*NOAH Compendium, 2023*; *National Office of Animal Health, 2022*). Combination spot-on products containing imidacloprid and moxidectin (an anthelmintic parasiticide that is systemically absorbed following application) state that dogs should not be allowed to swim for 4 days after treatment, based on the aquatic toxicity of moxidectin. The Seresto collar® datasheet contains no warnings against swimming but states under 'Special precautions for disposal' that the product should not enter water courses as it may be dangerous for fish and other aquatic organisms. However, the manufacturer's website states that the product does not need to be removed when the dog goes swimming or is bathed (*Elanco, 2022*).

Datasheet guidelines on bathing following administration of a spot-on product are based on considerations of efficacy and vary between products. Minimum intervals before bathing following product application generally reflect guidelines for swimming. Frontline® (Boehringer-Inghelheim, containing fipronil) guidelines state that more frequent bathing than once a week should be avoided (*NOAH Compendium, 2023*), whereas the Advocate® (Elanco, containing imidacloprid and moxidectin) datasheet states more generally that 'Frequent shampooing or immersion in water after treatment may reduce the efficacy of the product' (*National Office of Animal Health, 2019*). Similarly, the datasheet for Seresto®

collars warns against extensive shampooing, without providing specific guidance—but states that monthly shampooing or water immersion does not significantly shorten the efficacy duration for ticks, with the product's flea efficacy gradually decreased from the 5th month.

Product datasheets for spot-on parasiticides generally advise owners not to stroke animals until the application site is dry. Guidance on other potential routes to the environment such as washing of bedding or disposal of excreta are not included in datasheet guidelines.

Most survey participants indicated that they were aware of and followed the relevant datasheet guidelines, with 84.4% (232/275) of dog owners who applied spot-ons or collars indicating that they were aware of product warnings regarding swimming and bathing after flea treatment.

## Activities affecting pathways to the environment

This survey focused primarily on the frequency of activities likely to result in DTD and direct transfer to waterways for ectoparasiticides administered to pets, namely bathing of dogs, washing of their bedding, swimming, and disposal of excreta. Spot-on fipronil has been demonstrated in bathing rinsate from treated dogs for at least 28 days after application (*Teerlink, Hernandez & Budd, 2017*), and spot-on ectoparasiticides have been demonstrated to persist and accumulate in pet bedding and the household environment (*Jacobs et al., 2001*; *Dyk et al., 2012*; *Mahler et al., 2009*). *Diepens et al. (2023)* identified swimming as a direct source of parasiticide emissions to waterways.

The frequency of both swimming and bathing activity in dogs was affected by spot-on application—likely a reflection of owner awareness of product guidelines on these activities. Spot-on treated dogs were reported to swim less frequently than non spot-on treated dogs, with 36.2% swimming at least monthly, compared to 49.7% of non-spot-on treated dogs. However, even with reduced swimming frequency in this population, we estimate that dogs treated with spot-on imidacloprid, banned for outdoor use in agriculture, may be entering UK waterways over 3 million times a year. Similarly, spot-on treated dogs were less likely to be bathed at frequent (weekly or more) intervals than untreated dogs, however 54.6% of spot-on treated dogs were reported to be bathed at least monthly, and we estimate that imidacloprid spot-on treated dogs are bathed over 5 million times each year in the UK. Most bathing of dogs (77.1%) was reported to occur in locations that drain to the sewage system, suggesting that most bathing events were likely to result in down-the-drain transfer.

Both cats and dogs were found to sleep on a variety of surfaces that are likely to be laundered, including family beds and their own bedding, and washing of pet bedding was not affected by ectoparasiticide treatment. 67.2% of dog owners reported washing their bedding at least monthly, and 87.8% of dogs had their bedding washed at least every 3 months. It is therefore likely that DTD ectoparasiticide washoff from bedding is likely to occur for the majority of the estimated 9.4 million imidacloprid spot-on doses applied to dogs annually. Cat owners reported less frequent bed washing, but 69.1% of cat owners reported washing their bedding at least every 3 months, indicating that this may also be a significant pathway to waterways for ectoparasiticides applied to cats.

A relatively small percentage of pet owners reported that they regularly disposed of animal waste by flushing it down the toilet (0.8% of dogs and <10.1% of cats), suggesting that this may be a comparatively minor pathway to waterways for parasiticides. However, over 70% of cat owners indicated that their cat always or sometimes toileted outside, and 34.8% of dog owners said that they sometimes or always failed to dispose of dog stool in a waste bin—supporting concerns that pet excreta is likely to be a route to the terrestrial environment for some parasiticides, particularly anthelmintics and isoxazoline ectoparasiticides such as fluralaner which are primarily excreted in the stool (*Merck Co., Inc., 2016*; *European Medicines Agency , 2013*).

Other likely pathways to waterways remain to be investigated. These include walking dogs in the rain, laundering of owner clothing, and owner bathing/showering. In particular, transfer of spot-on residue to hands has been confirmed (*Craig et al., 2005*; *Dyk et al., 2012*) and represents an opportunity for further research as a potential route of DTD emissions.

## CONCLUSION

Historically, the environmental impact of parasiticides used in companion animals has been assumed to be low, and little consideration has been given to this in past years. Extensive usage continues, and whilst the environmental impact of a parasiticide applied to an individual animal may be low or even insignificant, the potential cumulative impact of parasiticide emissions from many millions of pets treated multiple times each year is of serious concern.

Of particular concern is evidence that veterinary usage is contributing to widespread fipronil and imidacloprid waterway pollution, occurring at levels likely to impact aquatic ecosystems (*Perkins et al., 2020*). Confirmation of the ubiquitous presence of both fipronil and imidacloprid in wastewater effluent from multiple UK treatment plants (*UK Water Industry Research 2023*), supports concerns that DTD transfer from households is likely to be a significant source of pollution. Survey results reported here indicate that wash-off from millions of fipronil and imidacloprid-treated animals is likely to be passing DTD annually through bathing and washing of bedding, and several other likely pathways to waterways exist.

There is an urgent need for further research to investigate the environmental impact of companion animal parasiticides. This study is intended as a step towards quantifying the pathways to the environment through assessing the frequency of activities likely to result in transfer of ectoparasiticides to waterways. Further studies are needed to establish emissions fractions for identified routes, including bathing, swimming and washing of bedding, in order to quantify the ectoparasiticide load entering the environment and help inform regulatory and prescribing decisions.

## ACKNOWLEDGEMENTS

The authors would like to thank the survey respondents who generously shared their time, and Martin Whitehead for his assistance with the survey design and manuscript.

### Funding

This work was supported by the UK Veterinary Medicines Directorate (VMD). The VMD (Veterinary Medicines Directorate) assisted with the review of the draft manuscript. The funders had no role in study design, data collection and analysis, decision to publish, or preparation of the manuscript.

### Grant Disclosures

The following grant information was disclosed by the authors:
UK Veterinary Medicines Directorate.

### Competing Interests

The authors declare there are no competing interests.

### Author Contributions

- Rosemary Perkins conceived and designed the experiments, performed the experiments, analyzed the data, prepared figures and/or tables, authored or reviewed drafts of the article, and approved the final draft.
- Dave Goulson conceived and designed the experiments, authored or reviewed drafts of the article, and approved the final draft.

### Data Availability

The survey responses are available in the Supplemental Files.

### Supplemental Information

Supplemental information for this article can be found online at http://dx.doi.org/10.7717/peerj.15561#supplemental-information.

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
