# Peer review of "To flea or not to flea: survey of UK companion animal ectoparasiticide usage and activities affecting pathways to the environment"

_PeerJ, doi:10.7717/peerj.15561_

## Round 0.1 · original submission · Minor Revisions

Dear Authors,

After their scrutiny, the referees suggest your submission deserves minor revisions prior to its acceptance.

I hope you will apply the suggestions, and resubmit your manuscript as soon as possible.

Thank you very much,
Francesco

Reviewer 1 ·

Basic reporting

COMMENTS TO THE AUTHORS
The Manuscript 80021, entitled “To flea or not to flea: Survey of UK companion animal ectoparasiticide usage and activities affecting pathways to the environment” provides data on the effect these treatments have on the environment. Results presented in the manuscript are of interest. However, some revisions are needed before it become suitable for publication.

Lines 52-55: this is extremely reductive compared to the different compounds and formulation available. Pirethroids have not been reported for pets and, specifically, flumethrinfor cats. Other strategies for the control of ectos in pets are also important. See for example Otranto D, Wall R. New strategies for the control of arthropod vectors of disease in dogs and cats. Med Vet Entomol. 2008 Dec;22(4):291-302. doi: 10.1111/j.1365-2915.2008.00741.x. Epub 2008 Sep 8. PMID: 18785935.

Lines 99-100: Also here, authors should be clear about the reason for not including the pirethroids as ectopiricides.

Lines 113-115: Considering the aims of the study, all the information about other ectoparasiticides should be carefully reviewed.

Experimental design

Commented in the main text

Validity of the findings

Commented in the main text

Additional comments

Commented in the main text

·

Basic reporting

The article is very clear. The introduction material provides an incredibly thorough overview of flea and tick ectoparasiticide use with comprehensive reference to both the environmental fate literature and veterinary literature in this topic area. The introduction alone provides an excellent review, not otherwise available in the literature, for those working in this area. The survey design and presentation of results are clear and relevant.

Experimental design

It is crucial to understand the intricacies of ectoparasiticides use in order for researchers to evaluate both the true environmental impact from their use as well as the potential for mitigation. The research question and survey are well designed and fill a critical gap in this area. The number of participants is beyond adequate to fill this data need. The survey provides meaningful data on many pathways yet to be fully evaluated in the literature and thus fills a critical data gap extremely useful to determine if flea and tick treatment use results in substantial environmental impact from activities such as laundry and bathing.

Validity of the findings

Survey response size, characterization of results, and statistical tools are all robust and clear.

Additional comments

I have two recommendations prior to publication in PeerJ.

The first is to add a figure that presents data solely by active ingredient combing with potentially application type separated as oral or external. One of the main drivers of this research is to evaluate down-the-drain use and potential environmental impacts. Thus seeing a data presented in this way can help better articulate how active ingredients are in highest use with highest potential for DtD and off-site transport.

Second, while it is beyond the scope to fully cover alternative flea and tick treatment options, I think it is at least expanding slightly the paragraph on UK veterinarian recommendations to make it more clear the general direction that is recommended in place of routine prophalytic use (lines 365-374).

---

## Round 0.2 · accepted · Accept

Dear Authors,

It is a pleasure to accept your submission for publication on PeerJ. Referee opinions were homogenous in their positive evaluation, and I agree.

I hope to see a further submission from you in the Journal soon.

Francesco Porcelli

·

Basic reporting

No concerns.

Experimental design

no concerns

Validity of the findings

no concerns

Additional comments

Edits fully address concerns raised in initial review.